# Curvature Regularization to Prevent Distortion in Graph Embedding

**Hongbin Pei**[1,4,*]
peihb15@mails.jlu.edu.cn

**Bingzhe Wei**[2]
bwei6@illinois.edu

**Kevin Chen-Chuan Chang**[2,3]
kcchang@illinois.edu

**Chunxu Zhang**[1,4]
cxzhang19@mails.jlu.edu.cn

**Bo Yang**[1,4,†]
ybo@jlu.edu.cn

[1]College of Computer Science and Technology, Jilin University, China
[2]Department of Electrical and Computer Engineering, University of Illinois at Urbana-Champaign, USA
[3]Department of Computer Science, University of Illinois at Urbana-Champaign, USA
[4]Key Laboratory of Symbolic Computation and Knowledge Engineering of Ministry of Education, China

## Abstract

Recent research on graph embedding has achieved success in various applications. Most graph embedding methods preserve the proximity in a graph into a manifold in an embedding space. We argue an important but neglected problem about this proximity-preserving strategy: Graph topology patterns, while preserved well into an embedding manifold by preserving proximity, may distort in the ambient embedding Euclidean space, and hence to detect them becomes difficult for machine learning models. To address the problem, we propose *curvature regularization*, to enforce flatness for embedding manifolds, thereby preventing the distortion. We present a novel angle-based sectional curvature, termed *ABS curvature*, and accordingly three kinds of curvature regularization to induce flat embedding manifolds during graph embedding. We integrate curvature regularization into five popular proximity-preserving embedding methods, and empirical results in two applications show significant improvements on a wide range of open graph datasets.

## 1 Introduction

Recent research on graph embedding has achieved considerable success to represent graph data in various applications, such as node classification [1, 2], link prediction [3, 4], community discovery [5, 6], and recommendation [7, 8]. Graph embedding aims to encode the topology patterns in a graph into distributed vector representations, which can be readily fed into machine learning models for downstream applications. Thus, graph embedding bridges the gap between non-Euclidean graph structures and machine learning models operating in an Euclidean space.

Most graph embedding methods aim to preserve the proximity in a graph into an embedding space. For instance, the objective of node embedding is that proximal nodes in a graph should have similar representations in the embedding space, wherein proximal nodes are usually defined as co-occurrence nodes in a random walk path [9] or neighboring nodes some hops away [10]. Such proximity-preserving embedding methods are efficient and flexible, and the commonly employed models include matrix factorization, deep neural networks, and edge reconstruction models [11].

---

[*]This work is conducted during his visit at University of Illinois at Urbana-Champaign.
[†]Corresponding author.

As graph embedding places nodes on an embedding manifold (a metric space containing a set of points and a geodesic distance function) in an ambient embedding Euclidean space, we argue an important but neglected problem about the proximity-preserving graph embedding.

> *Graph topology patterns, while preserved well into an embedding manifold by preserving proximity, may distort in the ambient embedding Euclidean space, and hence to detect them becomes difficult for machine learning models that operate in an Euclidean space.*

**Motivating case study.** We present a case study to illustrate and analyze such pattern distortion, as shown in Figure 1. For a toy graph in panel A, we seek its 2-D node embedding by preserving proximity, as shown in panel B1. Meanwhile, an oracle node embedding is estimated through isometric embedding as a comparison in panel B2. The adopted embedding methods are Isomap [12] and its variant (see Appendix for technical details). We have three observations.

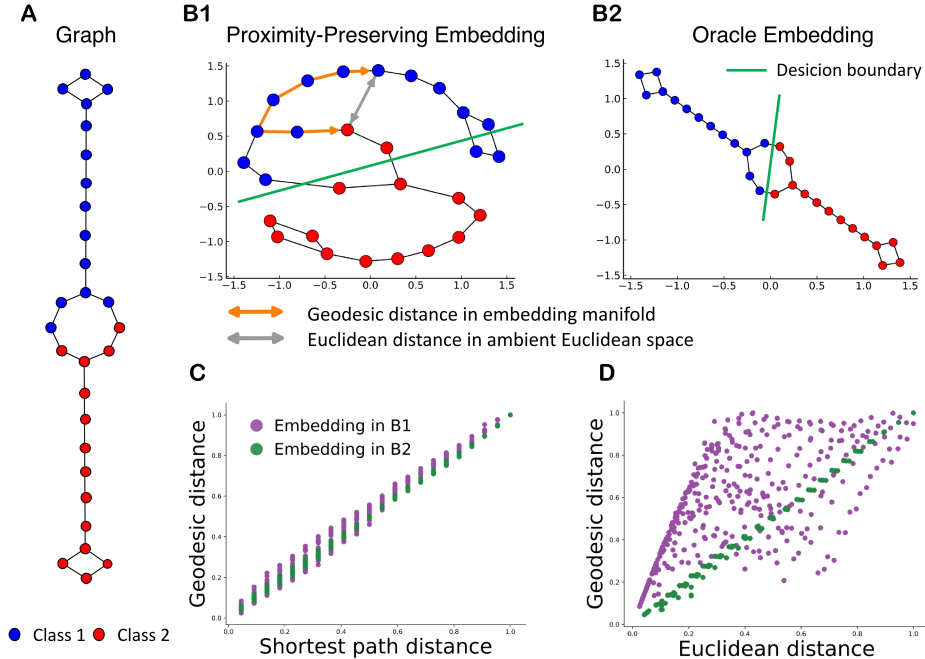

Figure 1: A case study on pattern distortion in graph embedding.

*Observation 1: Graph topology-patterns are preserved in an embedding manifold via proximity-preserving graph embedding.* For both B1 and B2 embeddings, the connected nodes in the graph tend to be close in the embedding space since the proximity in the graph is preserved. From the perspective of geometry, proximity-preserving graph embedding is a mapping from a graph to its isometric homeomorphic manifold [13]. In other words, for any two nodes, their shortest-path distance in the graph is consistent with their geodesic distance[3] in the embedding manifold. As shown in panel C, the two embeddings in B1 and B2 both preserve well the consistency between the two kinds of distance, so they both preserve graph proximity– and thus also the implied topology patterns– in their embedding manifolds via proximity-preserving embedding. E.g., the homophily in the graph (i.e., similar nodes are more likely to connect than dissimilar ones [14]) is presented as spatial clustering (the green boundary) of similar nodes, in term of node labels, in the embeddings in both B1 and B2.

*Observation 2: However, such patterns inevitably distort in the ambient embedding space.* For the embedding in panel B1, nodes far apart in embedding manifold, as measured by geodesic distance (the orange curve with arrows), may have a short straight-line distance (grey lines with arrows) in the ambient Euclidean space of the embedding manifold. As the embedding manifold may be folded, twisted, and curved in the ambient Euclidean space, such distance divergence is inevitable for embeddings learned by preserving proximity. As shown in panel D, the distance divergence is much larger in a curved embedding manifold (panel B1) than in a flat embedding manifold (panel B2). The

divergence between graph and Euclidean distances reflects that geometry patterns in the embedding manifold distort in its ambient Euclidean space.

Thus, we define pattern *distortion* as the distance divergence between an embedding manifold and its ambient Euclidean space. Mathematically, we say the manifold has distortion $\rho$ in the ambient Euclidean space,

$$\rho = \frac{1}{n(n-1)} \sum_{i \neq j} \frac{d_{\mathcal{M}}(\boldsymbol{x}_i, \boldsymbol{x}_j)}{d_{\mathcal{E}}(\boldsymbol{x}_i, \boldsymbol{x}_j)}, \tag{1}$$

where $n$ is the number of nodes, vector $\boldsymbol{x}_i$ and $\boldsymbol{x}_j$ are the representations of node $v_i$ and $v_j$ in the embedding space, and $d_{\mathcal{M}}(\cdot, \cdot)$ and $d_{\mathcal{E}}(\cdot, \cdot)$ denote the geodesic distance function in the embedding manifold and the Euclidean distance function in the ambient Euclidean space, respectively. The minimum of the distortion is 1.0, which indicates a flat manifold where geodesic distance is equal to Euclidean distance, e.g., a straight line.

*Observation 3: Such distortion hinders machine learning applications operating in an embedding space.* The embedding in B2 exhibits more discriminative spatial distribution than B1, which is desirable for machine learning applications, e.g., node classification, because of a large margin. Obviously, the difficulties to detect patterns from the two embeddings are very different, although both of them similarly preserve the graph topology patterns in their embedding manifolds (Observation 1). Since the geometry patterns preserved in an embedding manifold may distort in its ambient Euclidean space (Observation 2), which most of popular machine learning models [15] operate in and hence "see" only the Euclidean patterns, detecting those patterns becomes difficult or even impossible.

Now, we can conclude that a good embedding should not only preserve topology patterns in a graph (*Observation 1*), but also eliminate the distortion of its embedding manifold in the ambient Euclidean space (*Observation 2*), so as to present Euclidean patterns that can be detected effectively by machine learning models (*Observation 3*). This case study, although leveraging specific embedding methods (Isomap and its variant), is representative of proximity-preserving graph embedding methods, because such distortion is inevitable due to their common strategy of (only) proximity preservation. We note that many popular graph embedding methods (e.g., matrix factorization and deep learning-based methods) use dot product to measure the similarity between representations of nodes in an embedding space, which is different from Euclidean distance used in this case study. However, the distortion measure $\rho$ defined in Eq.1 also apply to reflect the pattern distortion for those embedding methods because dot product is highly related to Euclidean distance [16]. It's important to note that the pattern distortion not only exists in Euclidean embedding, but also exists in non-Euclidean embedding [17].

Existing proximity-preserving embedding methods mostly focus on how to preserve the topology patterns effectively and efficiently, but rarely notice the problem to eliminate the distortion, which motivates us to study this neglected but important problem. In the literature, a straightforward strategy to prevent the distortion is isometric embedding, e.g., Isomap, the method used to obtain the oracle embedding in Figure 1B2. However, isometric embedding involve scalability problem because of its very high time complexity, which limit their applications in practice.

**Our solution.** In this paper, we address the distortion problem in proximity-preserving graph embedding through an intuitive idea: *Since the distortion is caused by curved embedding manifolds, if we can enforce the embedding manifolds to be flat, the distortion will be prevented.* Fortunately, curvature, a concept from differential geometry [18], aims to measure how far a manifold (e.g., a curve or a surface) is from being flat. If we have one curvature that can apply to embedding manifold, we can restrict the curvature during proximity-preserving embedding, so as to induce a flat embedding manifold which has a low distortion $\rho$ in ambient Euclidean space.

The challenge to implement the idea is lacking a existing curvature that can measure embedding manifold in graph embedding setting. In the literature, most of curvatures, e.g., Gaussian curvature, are defined by the gradient of manifold function which cannot get in graph embedding setting where we only know the node/edge representations, i.e., a set of points, in embedding space. In this paper, we fill this void by proposing a novel angle based sectional curvature, termed ABS curvature, to measure the curvature of the embedding manifold. Accordingly, we present three kinds of curvature regularization that can be readily incorporated into existing proximity-preserving node embedding methods so as to induce flat embedding manifolds with a low distortion. Finally, we empirically validate the algorithm by incorporating the regularization term into five popular node embedding methods for performing node classification and link prediction tasks on eight open datasets of graphs.

In summary, the contribution of this paper is four-fold: 1) We raise and formulate an important but neglected problem in proximity-preserving graph embedding, the geometry patterns in embedding manifold distort in the ambient Euclidean space; 2) We provide a solution to prevent the distortion, to restrict the curvature of embedding manifold during embedding; 3) We propose a novel curvature, ABS curvature, for embedding manifold, present three kinds of curvature regularization, and develop a curvature regularization optimizing algorithm; 4) We validate our method by incorporating the regularization term into five popular node embedding methods on eight open datasets of graphs.

## 2    Problem Formulation

For convenience of presentation, this paper only considers the setting of proximity-preserving node embedding. The developed method can be readily extended to other graph embedding tasks, e.g., edge embedding. We firstly describe proximity-preserving node embedding from a geometry perspective.

Let $\mathcal{G} = (V, E)$ be a graph, where $V$ and $E$ are node set and edge set, respectively. Each edge $e_{i,j} \in E$ connects two nodes $v_i, v_j \in V$. Let $f : v_i \to \boldsymbol{x}_i$ be an embedding function mapping a node $v_i$ in graph to a representation vector $\boldsymbol{x}_i$ of $d$ dimension, $d \ll |V|$. The representation $\boldsymbol{x}_i \in X$ can be considered as a point in an embedding (Riemannian) manifold $\mathcal{M}$, and $X$ is a set of points. The geodesic polygonal curve in the manifold, like straight line in the Euclidean space, is denoted by $P_{i,j}$ and defined by the ordered nodes along the shortest paths in the graph. The length of geodesic polygonal curve is geodesic distance $d_{\mathcal{M}}$. The manifold is embedded in an ambient Euclidean space, and around any point $\boldsymbol{x}_i$, the manifold locally resembles a Euclidean tangent space $\mathcal{T}_{\boldsymbol{x}_i}\mathcal{M}$.

**Definition 1.** *(Geodesic Polygonal Curve) For two points $\boldsymbol{x}_i$ and $\boldsymbol{x}_j$ in embedding manifold $\mathcal{M}$ of graph $\mathcal{G}$, the geodesic polygonal curve $P_{i,j}$ between them is a polygonal curve specified by a sequence of edges $e_{q',q''}, (q', q'') \in \Gamma_{i,j}$, where $\Gamma_{i,j}$ is an ordered set and contains the indexs of edges along the shortest path between node $i$ and $j$ in graph $\mathcal{G}$.*

**Definition 2.** *(Geodesic Distance) Geodesic distance in embedding manifold $\mathcal{M}$ is the length of a geodesic polygonal curve. Specifically, for two points $\boldsymbol{x}_i$ and $\boldsymbol{x}_j$ in $\mathcal{M}$,*

$$d_{\mathcal{M}}(\boldsymbol{x}_i, \boldsymbol{x}_j) = \sum_{(q',q'') \in \Gamma_{i,j}} d_{\mathcal{E}}(\boldsymbol{x}_{q'}, \boldsymbol{x}_{q''}), \qquad (2)$$

*where $d_{\mathcal{E}}(\boldsymbol{x}_{q'}, \boldsymbol{x}_{q''})$ is the local Euclidean distance between the two consecutive points along $P_{i,j}$, the geodesic polygonal curve.*

For any node $v_i$, proximity-preserving graph embedding requires its neighbor node $v_j$ has a similar representation $\boldsymbol{x}_j$ with $\boldsymbol{x}_i$ in local Euclidean tangent space $\mathcal{T}_{\boldsymbol{x}_i}\mathcal{M}$. Specifically, the proximal nodes are commonly defined as co-occurrence nodes in a random walk path [9] or neighbor nodes one hop or $r$ hops away [10]. And the similarity in local tangent space is usually measured by Euclidean distance [19, 20] or dot product [5, 21]. The problem we study in this paper is how to minimize the distortion $\rho$ defined by Eq.1 during above proximity-preserving node embedding.

## 3    Preliminaries

**Curvature.** Curvature is a concept from differential geometry, and it measures how far a manifold (e.g., a curve or a surface) is from being flat. Taking curve as an example, we can intuitively consider curvature as the difference from a curve to its tangent at a point. The curvature of a straight line is zero. Mathematically, the curvature of a curve is defined as the rate of change of direction of tangent at a point that moves on the curve at a constant speed.

For 2-dimensional surface, normal curvature is usually used to measure how a surface differs from its flat tangent plane around one point. Consider planes that are perpendicular to the tangent plane at a point on the surface, i.e., the planes contain the normal vector of the surface at that point. Those planes are called normal plane. The intersection of a normal plane with the surface forms a normal section (i.e., a curve) whose curvature is able to profile the curvature of the surface along the curve. The curvatures of the normal sections formed by all normal planes are called normal curvatures of the surface at the point, as illustrated in Figure 2A. The maximum and minimum normal curvatures are called principal curvatures; the product of the two principal curvatures is the Gaussian curvature.

For $n$-dimensional manifold embedded in a ambient Euclidean space, *sectional curvature* is one way to define its curvature, which is an extension of normal curvature to high dimensional space.

Like surfaces have tangent planes, a $n$-dimensional manifold at a point also has a Euclidean tangent space of $n$-dimension. For a plane in the tangent space, there is a surface in the manifold, the surface has the plane as its tangent plane. That is, the surface consists of geodesic curves in the manifold emanating from the point through all the directions in the tangent plane. The curvatures of these geodesic curves reflect how the manifold differs from its tangent space along the directions. The Gaussian curvature of the surface is called the sectional curvature of the manifold for the tangent plane, and the sectional curvatures for all tangent planes can completely describe the curvature of the manifold around the tangent point.

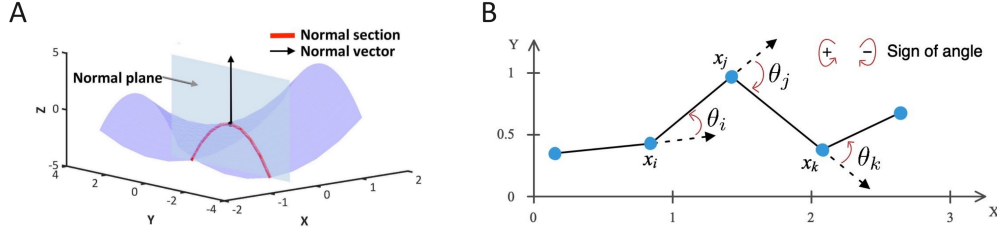

Figure 2: Panel A is an illustration of normal curvature of 2-dimensional surface. The intersection of a normal plane with the surface forms a normal section (red curve) whose curvature is used to profile the curvature of the surface along the curve. Panel B is an illustration of angle based curvature of polygonal curve. The angle based curvature at point $\boldsymbol{x}_i$ is defined as the turning angle, $\kappa_i = \theta_i$. Red arrows denote the sign of the angles.

**Discrete curvature.** All above curvatures are designed for continuous differentiable object. Now we describe *angle based curvature*, which aims to measure the curvature of discrete polygonal curve, e.g., the aforementioned geodesic polygonal curve in embedding manifold. Recall that curve curvature is modeled as the rate of change of direction of tangent at a point that moves on the curve. Extending this concept to polygonal curve, as shown in Figure 2B, the direction of tangent (dash arrow) of polygonal curve (black line) at point $\boldsymbol{x}_i$ changes by a signed angle $\theta_i$, i.e., the turning angle at point $\boldsymbol{x}_i$. In other words, the curvature of polygonal curve at a point can be measure by the turning angle, and hence the turning angle is called the angle based curvature. Mathematically, the angle based curvature at the point $\boldsymbol{x}_i$ is defined as $\kappa_i = \theta_i$.

## 4 Methods

To address the distortion problem, our basic idea is to restrict the curvature of embedding manifold during graph embedding, thereby inducing a flat embedding manifold with a low distortion $\rho$, Eq.1. In this section, we start by proposing a novel angle based sectional curvature for embedding manifold, and then accordingly present three kinds of curvature regularization that can be readily incorporated into existing proximity-preserving node embedding methods.

### 4.1 Angle-based Sectional Curvature

In the literature, there is no off-the-shelf curvature which can apply to embedding manifold. We proposed a novel angle based sectional curvature, termed ABS curvature, to fill the void. ABS curvature measures how an embedding manifold differ from its tangent space via the curvature of geodesic polygonal curve in embedding manifold. We design ABS curvature from two *basic assumptions*. The first is similar to the aforementioned sectional curvature, i.e., the curvature of geodesic curve can be used to reflect how the manifold differs from its tangent space along the direction of the curve. However, it's difficult to calculate the curvature of geodesic curve in graph embedding setting because we cannot obtain the functions of geodesic curves in embedding manifold. Thus, the second assumption we made is geodesic polygonal curve formed by points can approximate the geodesic curve in embedding manifold. We give a formal definition of ABS curvature.

**Definition 3.** *(Angle-based Sectional Curvature) For any geodesic polygonal curve $P_{i,j}$ that passes through a point $\boldsymbol{x}_q$ in embedding manifold $\mathcal{M}$, the angle-based sectional curvature at $\boldsymbol{x}_q$ along $P_{i,j}$, $K_q(P_{i,j})$, is the angle-based curvature of $P_{i,j}$ at $\boldsymbol{x}_q$,*

$$K_q(P_{i,j}) = \kappa_q, \tag{3}$$

*where $\kappa_q$ is defined on the geodesic polygonal curve $P_{i,j}$.*

The ABS curvature $K_q(P_{i,j})$ is defined on a geodesic polygonal curve $P_{i,j}$ (i.e., a section of embedding manifold), and is an extrinsic curvature. We then present the notations to denote the curvature of the entire embedding manifold. Curvature vector $K_q$ is formed by all curvatures $K_q(P_{i,j})$ defined on every geodesic polygonal curve passing through the point $\boldsymbol{x}_q$, which describes the curvature of $\mathcal{M}$ at $\boldsymbol{x}_q$. The curvature vectors $K_q$ at every point in $\mathcal{M}$ form a curvature vector field $\mathcal{K}$, which provides a completed description for the curvature of embedding manifold $\mathcal{M}$.

We then provide a connection between the proposed ABS curvature and the distortion to support our basic idea theoretically. Theorem 1 states that the distortion $\rho$ can be prevented by decreasing the absolute value of ABS curvature of embedding manifold under given conditions. Those conditions limit the theorem in the scope that ABS curvature can effects independently, since curvature and torsion can both influence the distortion in high-dimensional space [13]. To meet the conditions during graph embedding, we design specific optimization strategy in the following algorithm.

**Lemma 1.** *In a 2-dimensional embedding manifold $\mathcal{M}$, the distortion $\rho$ is an increasing function of $|K_q(P_{i,j})|$, the absolute value of ABS curvature at any $\boldsymbol{x}_q$ along $P_{i,j}$, if the absolute value of summation of curvatures along any part of $P_{i,j}$ is less than $\frac{\pi}{2}$, $|\sum_p K_p(P_{i,j}^s)| < \frac{\pi}{2}$. $P_{i,j}^s$ is a part of $P_{i,j}$ and $\boldsymbol{x}_p$ is a point along $P_{i,j}^s$.*

*Proof.* Please see the Appendix for proof. $\qquad\square$

**Theorem 1.** *In a n-dimensional embedding manifold $\mathcal{M}$, the distortion $\rho$ is an increasing function of $|K_q(P_{i,j})|$, the absolute value of ABS curvature at any $\boldsymbol{x}_q$ along $P_{i,j}$, if the absolute value of summation of curvatures along any part of $P_{i,j}'$ is less than $\frac{\pi}{2}$, $|\sum_p K_p(P_{i,j}'^s)| < \frac{\pi}{2}$. $P_{i,j}'$ is the projection of $P_{i,j}$ in the 2-dimensional subspace of embedding manifold $\mathcal{M}$, $P_{i,j}'^s$ is a part of $P_{i,j}'$ and $\boldsymbol{x}_p$ is a point along $P_{i,j}'^s$.*

*Proof.* Please see the Appendix for proof. $\qquad\square$

## 4.2 Curvature Regularization

In this section, we firstly present curvature regularization, a form of regularization based on the proposed ABS curvature. we then design two efficient variants of curvature regularization to solve the scalability of the curvature regularization. Finally, we develop an optimizing algorithm to optimize the curvature regularization during proximity-preserving graph embedding, so as to induce flat embedding manifolds.

From Theorem 1, we expect to decrease the absolute value of ABS curvature of embedding manifold. As it's difficult to optimize the absolute value of a angle directly, we design the curvature regularization as the cosine of the ABS curvature because the cosine value of a angle increases with the absolute value of the angle when the angle is in $[-\pi, \pi]$ and the cosine function is convex and easy to calculate through vector product. The curvature regularization is given by

$$\Omega_c(X) = \sum\nolimits_{K_q(P_{i,j}) \in \mathcal{K}} \cos(K_q(P_{i,j})). \tag{4}$$

To obtain the ABS curvature vector filed $\mathcal{K}$ involve the problem of scalability because it need to calculate the shortest path between every node pair in a graph. We adopt a sampling strategy to solve the problem, and the sampled curvature regularization is given by

$$\Omega_s(X) = \sum\nolimits_{v_i, v_j \in S} \cos(K_q(P_{i,j})), \tag{5}$$

where $S$ is a set of sampled nodes. The sampled curvature regularization only restricts the ABS curvature along geodesic polygonal curve $P_{i,j}$ between sampled nodes, $v_i, v_j \in S$. A comprehensive and unbiased sampling is crucial for this regularization.

In real-world applications, random walk-based embedding methods are usually leveraged to deal with very large graphs in which it is also very computationally difficult to obtain sampled shortest paths. We design a approximated curvature regularization for random walk-based embedding methods specifically.

$$\Omega_a(X) = \sum\nolimits_{r \in R} \cos(K_q(r)), \tag{6}$$

where $r$ is a acyclic path generated by random walk in embedding method, and $R$ is a set of random-walk paths. In this regularization, we replace shortest paths by random-walk paths because random-walk paths may include lots of shortest paths in graph.

Based on curvature regularization, we propose an algorithm to optimize the curvature regularization during proximity-preserving graph embedding, as shown in Algorithm 1. There are two phases in the algorithm. In first phase, we minimize the embedding loss term $\mathcal{L}(\mathcal{G})$ and the curvature regularization term $\Omega(X)$ separately, so as to obtain an embedding manifold with low enough ABS curvature, thereby making most of geodesic polygonal curves meet the conditions in the theorem 1. In second phase, we minimize the two terms jointly to get a flat embedding manifold which preserves well the proximity in graph and has a low distortion $\rho$. Weight $\lambda$ is a trade-off hyperparameter.

---

**Algorithm 1** Curvature regularization optimizing algorithm.

---

1: **input:** graph $\mathcal{G}$
2: **output:** node representations $X$
3: **preprocessing:** get paths in $\mathcal{G}$        ▷ Shortest paths ($\Omega_c$ and $\Omega_s$) or random-walk paths ($\Omega_a$)
4: **for** $t$ iterations **do**                                                  ▷ **First phase**
5:      **while** not converged **do**
6:          minimize embedding loss term $\mathcal{L}(\mathcal{G})$
7:      **while** not converged **do**
8:          minimize curvature regularization term $\Omega(X)$        ▷ Specific term is $\Omega_c$, $\Omega_s$, or $\Omega_a$
9: **while** not converged **do**                                     ▷ **Second phase**
10:      minimize the two terms jointly $\mathcal{L}(\mathcal{G}) + \lambda\Omega(\boldsymbol{X})$      ▷ $\lambda$ is a trade-off hyperparameter

---

## 5 Experiments

We comprehensively validate our method on both node classification (NC) and link prediction (LP) tasks, on eight open graph datasets. We integrate curvature regularization into five popular proximity-preserving node embedding methods, and then compare performance of the embedding methods with the curvature regularization against the original embedding methods, and empirically demonstrate significant improvements.

### 5.1 Experimental Setup

**Datasets**. We evaluate the proposed method on eight open graph datasets described below (more details are available in the Appendix). In all experiments, we do not use node attribute information, and only use node representations learned by node embedding methods. Empirical results show significant improvements on a wide range of open graph datasets.

(1) *Citation networks*. Cora, Citeseer and Pubmed are citation network benchmark datasets [22, 23], where nodes are papers and edges are citation links. Node labels are the academic topic of papers.
(2) *WebKB*. WebKB contains a subset of web pages collected from computer science departments. Specifically, we use the subgraphs from four universities, i.e., Cornell, Texas, Washington and Wisconsin. In these networks, nodes are web pages and edges are hyperlinks between pages. The web pages are classified into the five categories, student, project, course, staff, and faculty.
(3) *Polblogs*. Political blog network [24] consists of blogs about US politics and the links between them. Blogs are divided into two communities based on their political labels (liberal and conservative).

**Proximity-preserving node embedding methods**. We integrate curvature regularization into the following five popular node embedding methods to learn node representations. *Matrix Factorization (MF)* represents graph property with a matrix, and it can factorize this matrix to obtain low-dimensional vector representations of nodes [25]. *Laplacian Eigenmaps (LE)* learns node representations by factorizing graph laplacian eigenmaps [19]. *DeepWalk* learns node representations from random walk paths by leveraging Skip-Gram neural networks [9]. *Node2vec* is an extension of DeepWalk which adopts biased random walk and Skip-Gram neural networks to learn node representations [21]. *Structural Deep Network Embedding (SDNE)* learns node representations that preserve the first-order and second-order proximity with a deep autoencoder network [10].

**Parameter search**. For all node embedding models, we perform a random sampling hyper-parameter search on validation set of each dataset to get competitor models (See detailed hyper-parameter setting in the Appendix). The hyper-parameters searched over include the dimension of node representation as well as hyper-parameters specific to each model. We then integrate the curvature regularization term into those competitors, and only adjust the number of iteration $t$ and the weight $\lambda$ in algorithm.

## 5.2 Experimental Results

**Node classification**. With the node representations learned by each embedding method, we split randomly $60\%$ of the nodes in a graph as the training set and the remaining $40\%$ of nodes in a graph as the test set. Then we use a one-vs-rest logistic regression classifier to predict the labels. Accuracy is adopted as the evaluation metric in node classification task. We repeat the above process 10 times and average these results. Final results are summarized in Table 1. The reported numbers denote the mean classification accuracy in percent. We use suffix '-c', '-s', or '-a' to respectively denote $\Omega_c$, $\Omega_s$, or $\Omega_a$, the regularization term used. In general, curvature regularization improves the performance of those node embedding methods. The best performing method is highlighted in bold.

Table 1: Mean Classification Accuracy (Percent)

| Dataset | Cora | Cite. | Pubm. | Polb. | Wash. | Corn. | Texa. | Wisc. |
|---|---|---|---|---|---|---|---|---|
| MF | 44.30 | 54.11 | 39.16 | 51.10 | 55.07 | 43.60 | 28.53 | 20.38 |
| MF-c | 45.47 | 54.25 | **40.47** | 52.03 | 55.62 | **47.50** | **28.71** | **21.89** |
| MF-s | **48.37** | **54.38** | 40.05 | **55.58** | **56.44** | 46.20 | 28.70 | 21.71 |
| LE | 44.30 | 54.11 | 46.40 | 50.86 | 39.52 | 54.88 | 30.34 | 20.73 |
| LE-c | **45.80** | **54.52** | **48.37** | 51.02 | **45.71** | 55.12 | **31.61** | **24.31** |
| LE-s | 45.10 | 54.38 | 46.86 | **51.39** | 40.19 | **56.53** | 31.55 | 21.12 |
| SDNE | 42.07 | 53.35 | 39.43 | **78.13** | 30.34 | 26.43 | 50.14 | 43.56 |
| SDNE-c | **46.40** | **55.84** | 40.96 | 76.24 | **32.24** | **27.53** | 54.81 | **46.44** |
| SDNE-s | 44.63 | 50.84 | **41.47** | 76.79 | 30.49 | 24.53 | **55.32** | 44.23 |
| DeepWalk | 54.65 | 42.24 | 80.32 | 76.31 | 42.27 | 39.78 | 45.23 | 73.43 |
| DeepWalk-a | **62.05** | **53.42** | **80.43** | **79.34** | **46.18** | **42.31** | **47.79** | **76.89** |
| Node2vec | 48.02 | 62.29 | 44.25 | 93.72 | 48.86 | 69.18 | 54.79 | 42.50 |
| Node2vec-a | **52.91** | **66.59** | **46.16** | **94.60** | **51.31** | **73.05** | **60.27** | **46.30** |

**Link prediction**. In the link prediction task, we randomly remove $40\%$ of the links in the graph and then learn node representations from the subgraph induced from the remaining links. In order to predict the unobserved links, we need to construct link representations. Specifically, here we use link representations computed from the Hadamard product of the node representations of the nodes connected by a link. We adopt mean average precision to evaluate the performance, which is used to summarise precision-recall curves [26].

Table 2: Mean Average Precision (Percent)

| Dataset | Cora | Cite. | Pubm. | Polb. | Wash. | Corn. | Texa. | Wisc. |
|---|---|---|---|---|---|---|---|---|
| MF | 47.73 | 57.94 | 50.49 | 62.15 | 50.00 | 52.34 | 50.21 | 50.17 |
| MF-c | **49.83** | **65.88** | **51.98** | 63.43 | **57.41** | **57.30** | **52.10** | **50.56** |
| MF-s | 49.17 | 64.26 | 51.38 | **64.88** | 54.24 | 52.42 | 51.47 | 50.49 |
| LE | 49.82 | 52.14 | **50.96** | 50.51 | 80.26 | 70.28 | 61.57 | 68.18 |
| LE-c | **54.18** | **53.63** | 50.26 | 58.40 | **85.23** | **74.94** | **65.02** | **72.43** |
| LE-s | 53.15 | 51.50 | 50.12 | **62.13** | 77.17 | 72.12 | 57.43 | 69.17 |
| SDNE | 74.65 | 44.21 | 81.35 | 86.32 | 80.32 | 73.53 | 58.11 | 84.44 |
| SDNE-c | **78.78** | **48.53** | **85.31** | **90.23** | **84.97** | **79.45** | **85.86** | 82.94 |
| SDNE-s | 75.25 | 44.24 | 84.19 | 89.77 | 81.12 | 72.23 | 81.42 | 80.03 |
| DeepWalk | 53.58 | 57.84 | 59.11 | 63.21 | 94.46 | 88.78 | 59.29 | 83.75 |
| DeepWalk-a | **58.92** | **72.97** | **70.23** | **65.28** | **96.61** | **92.78** | **75.34** | **90.81** |
| Node2vec | 66.18 | 55.93 | 45.64 | 74.10 | 50.51 | 49.80 | 82.74 | 82.80 |
| Node2vec-a | **72.05** | **61.16** | **52.21** | **80.55** | **59.65** | **58.54** | **91.91** | **92.44** |

**Analysis of convergence and distortion reduction**. We conduct comparison experiments to analyze the convergence of curvature loss and the deduction of distortion (Eq. 1) during optimizing on the Cora dataset, as shown in the Appendix.

## 6 Discussions

In this section, we first discuss what are "good" curvatures and "bad" curvatures in graph embedding, then review the negative sampling strategy from the perspective of distortion prevention, and finally connect the curvature regularization to the betweenness centrality of graph.

## 6.1 "Good" Curvatures and "Bad" Curvatures

Curving an embedding space is indeed required for graph embedding, where a useful curvature is of benefit to preserving graph topology. For instance, the circles in Figure 1B2 indicate useful curvatures that form good node distributions, where the connected nodes are close and disconnected nodes are far apart. Hyperbolic space is also an example, whose useful negative curvature is suitable for embedding tree-structured graphs [17]. However, useless or harmful curvatures are also inevitable in proximity-preserving embedding because the existing strategies have no restriction on the "non-local" curves of an embedding space, such as the global "swirling" curve in Figure 1B1. Such harmful curvatures bring undesired distortions because unsupervised graph embedding aims to faithfully preserve graph topology patterns into an embedding space. In other words, any changes/distortions in patterns is not desired. We stress that the proposed algorithm differentiates useful from harmful curvatures because the former will contribute to the reconstruction term in the objective function while the latter will not and thus be eliminated by the curvature regularization during optimizing.

## 6.2 Reviewing Negative Sampling from the Perspective of Distortion Prevention

Negative sampling is an approximation strategy to alleviate the computational problem of Skip-Gram neural network [27] and it has been widely used in graph embedding methods [9]. In node embedding methods, the objective of negative sampling is to enlarge the distance between two disconnected nodes in Euclidean embedding space. From distortion perspective, negative sampling is one way to decrease the distortion of embedding manifold. Suppose node $v_j$ is a negative sample of $v_i$ and the geodesic distance between them $d_{\mathcal{M}}(\boldsymbol{x}_i, \boldsymbol{x}_j)$ in embedding manifold is fixed, if the Euclidean distance between them $d_{\mathcal{E}}(\boldsymbol{x}_i, \boldsymbol{x}_j)$ in ambient Euclidean space is enlarged by negative sampling, the term $\frac{d_{\mathcal{M}}(\boldsymbol{x}_i, \boldsymbol{x}_j)}{d_{\mathcal{E}}(\boldsymbol{x}_i, \boldsymbol{x}_j)}$ in distortion $\rho$, Eq.1, will decrease. However, negative sampling strategy may destroy preserving proximity in graph. If the sampled node is a proximal node to the target node, to enlarge the Euclidean distance between them will increase inevitably their geodesic distance and influence proximity preservation. Besides, negative sampling cannot guarantee the decrease of the distortion $\rho$ because to decrease the terms in $\rho$ corresponding to the sampled nodes may increase the other terms in $\rho$. Therefore, good negative samplers are very crucial for negative sampling strategy [28].

## 6.3 A Connection to Betweenness Centrality of Graph

In graph theory, betweenness centrality is a measure of centrality in a graph, which is defined as the number of shortest paths that pass through a node. The proposed curvature regularization is based on shortest paths and constrains more on nodes with high betweenness than others– which we think makes sense to prevent distortion: A node with high betweenness in a graph is corresponding to a "bottleneck" region in an embedding manifold, i.e., a small region where lots of geodesic curves pass through. Taking Figure 1B2 as an example, the nodes in the middle have high betweenness in graph. If we curve or fold the embedding space around these nodes, the Euclidean distances of most node pairs will be changed, which will bring large distortions in terms of Eq. 1, i.e., the divergence between the geodesic and Euclidean metric. To contrast, if we curve the embedding space around the nodes with low betweenness, it brings limited distortion. Thus, the curvature around high-betweenness nodes should be constrained more due to their high influence on distortion.

# 7 Conclusion

For the first time, we raised and formulated the pattern distortion problem in proximity-preserving graph embedding, which is important but neglected in existing works. As the distortion is caused by the curved embedding manifold, our basic idea to address the problem is to leverage *curvature regularization* to enforce flatness for embedding manifolds, thereby preventing the distortion. We proposed a novel angle-based sectional curvature, termed *ABS curvature*, for embedding manifold, and accordingly presented a curvature regularization to induce flat embedding manifolds during graph embedding. We further designed two efficient variants of curvature regularization to solve the scalability of curvature regularization. We developed an optimizing algorithm to induce an embedding manifold with low distortion during graph embedding. Finally, we evaluate the proposed algorithm by integrating curvature regularization into five popular proximity-preserving embedding methods. Comparisons on the NC and LP tasks show significant improvements on eight open graph datasets. As future work, we will attempt to apply the curvature regularization to more applications of graph embedding, such as graph alignment [29], community detection [30], structural comprehension [31, 32], and epidemic dynamic prediction [33, 34].

# 8 Broader Impact

This work raised the distortion problem in graph embedding for the first time. As the problem is important but neglected in the existing works, it will attract many researchers to design new approaches to address it. In this work, three kinds of curvature regularization and an optimizing algorithm have been proposed to prevent the pattern distortion during graph embedding. Obviously, the results of the work will have an immediate impact on improving the performance of various proximity-preserving graph embedding methods. This work will also benefit graph analysis applications in the real world, such as social network analysis, recommendation system, and knowledge graph mining.

The proposed models and algorithm advance the development of graph representation models which may bring negative societal consequences including privacy leak and fairness issues. For example, sensitive personal information, e.g., political orientation, occupation, and disease, may encode implicitly in user connections in social network. Those privacy can be learned effectively by graph representation models, and then may be leaked illegally by someone with bad intentions. Meanwhile, the links in social network also encode the information about population subgroups, such as gender and ethnic group. If such information is extracted by graph representation models and fed into downstream machine learning models, the trained model may lead to unfair predictions. For example, if one company scarcely employees women, models trained on this data would prefer man.

## Acknowledgments

We are very grateful to the anonymous reviewers for their constructive comments. This work was supported by the National Natural Science Foundation of China under grant 61876069; Jilin Province Key Scientific and Technological Research and Development Project under Grant Nos. 20180201067GX and 20180201044GX; Jilin Province Natural Science Foundation under Grant No. 20200201036JC; National Science Foundation IIS 16-19302 and IIS 16-33755; Zhejiang University ZJU Research 083650; Futurewei Technologies HF2017060011 and 094013; UIUC OVCR CCIL Planning Grant 434S34; UIUC CSBS Small Grant 434C8U; and IBM-Illinois Center for Cognitive Computing Systems Research (C3SR); and China Scholarships Council under scholarship 201806170202. Any opinions, findings, and conclusions or recommendations expressed in this publication are those of the author(s) and do not necessarily reflect the views of the funding agencies.

## Footnotes

[3]The geodesic distance between two nodes in an embedding manifold is defined as the length of the shortest polygonal curve between the two nodes (points). A shortest polygonal curve is shown as the orange curve in Figure 1 B1. A formal definition is in problem formulation section.

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
