[Supplementary Material]

# Appendix to the paper "Curvature Regularization to Prevent Distortion in Graph Embedding"

**Hongbin Pei**[1,4]
peihb15@mails.jlu.edu.cn

**Bingzhe Wei**[2]
bwei6@illinois.edu

**Kevin Chen-Chuan Chang**[2,3]
kcchang@illinois.edu

**Chunxu Zhang**[1,4]
cxzhang19@mails.jlu.edu.cn

**Bo Yang**[1,4]
ybo@jlu.edu.cn

[1]College of Computer Science and Technology, Jilin University, China
[2]Department of Electrical and Computer Engineering, University of Illinois at Urbana-Champaign, USA
[3]Department of Computer Science, University of Illinois at Urbana-Champaign, USA
[4]Key Laboratory of Symbolic Computation and Knowledge Engineering of Ministry of Education, China

## 1    Embedding Methods in Motivating Case Study

In the main text, we present a case study to illustrate and analyze pattern distortion in graph embedding. The oracle, and proximity-preserving embedding are obtained by Isomap [1] and its variant, respectively. Isomap is a nonlinear dimensionality reduction method and finds low-dimensional embedding of high-dimensional data by preserving the pairwise geodesic distances between data points in manifold.

To get the oracle embedding by Isomap, we first compute the length of shortest path between nodes in graph. Then, the node embedding is obtained by applying Multidimensional scaling (MDS) [2]. Particularly, the objective of Isomap is given by,

$$\min_{\boldsymbol{x}_i,\dots,\boldsymbol{x}_n} = \sum_{i>j}(\|\boldsymbol{x}_i - \boldsymbol{x}_j\| - d_{\mathcal{G}}(\boldsymbol{x}_i,\boldsymbol{x}_j))^2, \tag{1}$$

where $n$ is the number of nodes, vector $\boldsymbol{x}_i$ and $\boldsymbol{x}_j$ are the representations of node $v_i$ and $v_j$ in the embedding space, and $d_{\mathcal{G}}(\boldsymbol{x}_i,\boldsymbol{x}_j)$ denotes the length of shortest path between the two nodes. Isomap is an isometric embedding method since it preserves the distances between all nodes in graph.

We modify slightly Isomap to get the proximity-preserving embedding in motivating case. Here, we only preserve proximity in graph (the distances between two connected nodes) into embedding space. Thus, the objective becomes

$$\min_{\boldsymbol{x}_i,\dots,\boldsymbol{x}_n} = \sum_{e_{i,j} \in E}(\|\boldsymbol{x}_i - \boldsymbol{x}_j\| - d_{\mathcal{G}}(\boldsymbol{x}_i,\boldsymbol{x}_j))^2, \tag{2}$$

where $e_{i,j}$ denotes the edge between node $v_i$ and $v_j$, $E$ is the set of edge. As node $v_i$ and $v_j$ are connected, their distance $d_{\mathcal{G}}(\boldsymbol{x}_i,\boldsymbol{x}_j)$ in graph is set as one.

## 2    Proofs of Results

**Lemma 1.** *In a 2-dimensional embedding manifold $\mathcal{M}$, the distortion $\rho$ is an increasing function of $|K_q(P_{i,j})|$, the absolute value of ABS curvature at any $\boldsymbol{x}_q$ along $P_{i,j}$, if the absolute value of summation of curvatures along any part of $P_{i,j}$ is less than $\frac{\pi}{2}$, $|\sum_p K_p(P_{i,j}^s)| < \frac{\pi}{2}$. $P_{i,j}^s$ is a part of $P_{i,j}$ and $\boldsymbol{x}_p$ is a point along $P_{i,j}^s$.*

*Proof.* In the distortion $\rho = \frac{1}{n(n-1)} \sum_{i \neq j} \frac{d_{\mathcal{M}}(\boldsymbol{x}_i, \boldsymbol{x}_j)}{d_{\mathcal{E}}(\boldsymbol{x}_i, \boldsymbol{x}_j)}$, the geodesic distances between nodes $d_{\mathcal{M}}(\boldsymbol{x}_i, \boldsymbol{x}_j)$ in embedding manifold are determined and fixed by proximity-preserving embedding. Obviously, $\rho$ is a decreasing function of Euclidean distance $d_{\mathcal{E}}(\boldsymbol{x}_i, \boldsymbol{x}_j)$. We will proof the lemma by proofing Euclidean distance $d_{\mathcal{E}}(\boldsymbol{x}_i, \boldsymbol{x}_j)$ is a decreasing function of $|K_q(P_{i,j})|$.

In 2-dimensional embedding manifold $\mathcal{M}$, the geodesic polygonal curve $P_{i,j}$ can be projected on the straight line connected it two endpoints. Every line segment of $P_{i,j}$ has a corresponding line segment in the the straight line. An illustration is shown in Figure 1. There is a relationship between the length of the two kinds of line segment. $d_{\mathcal{E}}(\boldsymbol{x}_i, \boldsymbol{x}_k') = d_{\mathcal{E}}(\boldsymbol{x}_i, \boldsymbol{x}_k)\cos(\alpha_0)$, $d_{\mathcal{E}}(\boldsymbol{x}_k', \boldsymbol{x}_l') = d_{\mathcal{E}}(\boldsymbol{x}_k, \boldsymbol{x}_l)\cos(\alpha_0 + \theta_k)$, and for any line segment $d_{\mathcal{E}}(\boldsymbol{x}_{q'}', \boldsymbol{x}_{q''}') = d_{\mathcal{E}}(\boldsymbol{x}_{q'}, \boldsymbol{x}_{q''})\cos(\alpha_0 + \sum_{(\hat{q},\cdot) \in \Gamma_{i,j}, (\hat{q},\cdot) \leq (q',q'')} \theta_{\hat{q}})$ where $(\hat{q},\cdot)$ denotes an edge index before $(q', q'')$ in the order set $\Gamma_{i,j}$ (See definition of $\Gamma_{i,j}$ in Definition 1 in main text), and $d_{\mathcal{E}}(\boldsymbol{x}_{q'}, \boldsymbol{x}_{q''})$ is the local Euclidean distance between the two consecutive points along $P_{i,j}$.

Then we have the connection between Euclidean distance $d_{\mathcal{E}}(\boldsymbol{x}_i, \boldsymbol{x}_j)$ and the ABS curvature along the geodesic polygonal curve $P_{i,j}$,

$$d_{\mathcal{E}}(\boldsymbol{x}_i, \boldsymbol{x}_j) = \sum_{(q', q'') \in \Gamma_{i,j}} d_{\mathcal{E}}(\boldsymbol{x}_{q'}, \boldsymbol{x}_{q''})\cos(\alpha_0 + \sum_{(\hat{q},\cdot) \in \Gamma_{i,j}, (\hat{q},\cdot) \leq (q',q'')} \theta_{\hat{q}}). \qquad (3)$$

In the summation, the term $d_{\mathcal{E}}(\boldsymbol{x}_{q'}, \boldsymbol{x}_{q''})$ is determined and fixed by proximity-preserving embedding. As the absolute value of summation of curvatures along any part of $P_{i,j}$ is less than $\frac{\pi}{2}$, the cosine function is positive and the Euclidean distance is a decreasing function of the absolute value of any turning angle $\theta_{\hat{q}}$, i.e., $|K_q(P_{i,j})|$ the absolute value of ABS curvature at any $\boldsymbol{x}_q$ along $P_{i,j}$.

$\square$

Figure 1: An illustration of projection of geodesic polygonal curve. Green line denotes the straight line connected the two endpoints of geodesic polygonal curve $P_{i,j}$. After projecting the curve on the straight line, every line segment of the curve has a corresponding line segment in the straight line.

**Theorem 1.** *In a n-dimensional embedding manifold $\mathcal{M}$, the distortion $\rho$ is an increasing function of $|K_q(P_{i,j})|$, the absolute value of ABS curvature at any $\boldsymbol{x}_q$ along $P_{i,j}$, if the absolute value of summation of curvatures along any part of $P_{i,j}'$ is less than $\frac{\pi}{2}$, $|\sum_p K_p(P_{i,j}'^s)| < \frac{\pi}{2}$. $P_{i,j}'$ is the projection of $P_{i,j}$ in the 2-dimensional subspace of embedding manifold $\mathcal{M}$, $P_{i,j}'^s$ is a part of $P_{i,j}'$ and $\boldsymbol{x}_p$ is a point along $P_{i,j}'^s$.*

*Proof.* The Euclidean distance $d_{\mathcal{E}}(\boldsymbol{x}_i, \boldsymbol{x}_j)$ in n-dimensional space is a affine function of the Euclidean distance $d_{\mathcal{E}}(\boldsymbol{x}_i', \boldsymbol{x}_j')$ between the projection points ($\boldsymbol{x}_i'$ and $\boldsymbol{x}_j'$ are the projection points of $\boldsymbol{x}_i$ and $\boldsymbol{x}_j$) in the 2-dimensional subspace; thus $d_{\mathcal{E}}(\boldsymbol{x}_i', \boldsymbol{x}_j')$ is proportional to $d_{\mathcal{E}}(\boldsymbol{x}_i, \boldsymbol{x}_j)$. In the distortion $\rho = \frac{1}{n(n-1)} \sum_{i \neq j} \frac{d_{\mathcal{M}}(\boldsymbol{x}_i, \boldsymbol{x}_j)}{d_{\mathcal{E}}(\boldsymbol{x}_i, \boldsymbol{x}_j)}$, the geodesic distances between nodes $d_{\mathcal{M}}(\boldsymbol{x}_i, \boldsymbol{x}_j)$ in embedding manifold are determined and fixed by proximity-preserving embedding. Thus, $\rho$ in n-dimensional space is a decreasing function of Euclidean distance $d_{\mathcal{E}}(\boldsymbol{x}_i, \boldsymbol{x}_j)$ as well as $d_{\mathcal{E}}(\boldsymbol{x}_i', \boldsymbol{x}_j')$. In lemma 1, we have proved the Euclidean distance is a decreasing function of the absolute value of ABS

curvature in 2-dimensional space. Thus, $\rho$ in n-dimensional space is a increasing function of the absolute value of ABS curvature in 2-dimensional subspace.

The angle formed by two lines in n-dimensional space is also a affine function of the angle formed by the projections the lines in the 2-dimensional subspace. As the proposed ABS curvature is defined by angle, the absolute value of ABS curvature along a geodesic polygonal curve in n-dimensional space, $|K_q(P_{i,j})|$ , is proportional to the absolute value of ABS curvature along the projection of $P_{i,j}$ in 2-dimensional subspace, $|K_{q'}(P'_{i,j})|$. $P'_{i,j}$ is the projection of $P_{i,j}$ in the 2-dimensional subspace, and $\boldsymbol{x}_{q'}$ is a point on $P'_{i,j}$ Thus, we have the conclusion that $\rho$ in n-dimensional space is a increasing function of the absolute value of ABS curvature in n-dimensional subspace.                             □

## 3   Experimental Details

### 3.1   Dataset statistics

We detail the dataset statistics in Table 1.

Table 1: Statistics of Datasets.

| DATASET | CORA | CITE. | PUBM. | CORN. | TEXA. | WASH. | WISC. | POLB. |
|---------|------|-------|-------|-------|-------|-------|-------|-------|
| NODES   | 2708 | 3327  | 19717 | 183   | 183   | 230   | 251   | 1224  |
| EDGES   | 5429 | 4732  | 44338 | 295   | 309   | 446   | 499   | 16715 |
| CLASSES | 7    | 6     | 3     | 5     | 5     | 5     | 5     | 2     |

## 4   Training details

For all node embedding models, we perform a random sampling hyper-parameter search on validation set of each dataset to get competitor models. The hyper-parameters searched over include the dimension of node representation as well as hyper-parameters specific to each model. We then integrate the curvature regularization term into those competitors, and only adjust the number of iteration $t$ and the weight $\lambda$ in algorithm 1. We detail the hyper-parameter setting in Table 2,3,4,5,6 where weight denotes the weight of curvature regularization. In all experiments, we perform $t = 5$ iterations in first phase of algorithm 1. In random walk-based methods, we employ negative sampling strategy and negative sample number is set as $3$ or $5$.

Table 2: Parameter Setting for DeepWalk

| DATASET   | DIMENSION | WALKS LENGTH | WINDOW SIZE | WALKS NUM | WEIGHT |
|-----------|-----------|--------------|-------------|-----------|--------|
| CORA      | 128       | 64           | 3           | 32        | 10     |
| CITESEER  | 32        | 64           | 5           | 32        | 1      |
| PUBMED    | 64        | 64           | 5           | 32        | 0.001  |
| WEBKB     | 32        | 32           | 5           | 8         | 1      |
| POLBLOGS  | 32        | 64           | 5           | 32        | 0.001  |

Table 3: Parameter Setting for Node2vec

| DATASET   | DIMENSION | WALKS LENGTH | WINDOW SIZE | WALKS NUM | P   | Q   | WEIGHT |
|-----------|-----------|--------------|-------------|-----------|-----|-----|--------|
| CORA      | 128       | 64           | 5           | 32        | 0.5 | 0.1 | 10.0   |
| CITESEER  | 32        | 32           | 1           | 32        | 0.1 | 0.5 | 1.0    |
| PUBMED    | 64        | 64           | 1           | 128       | 0.9 | 0.1 | 1E-5   |
| WEBKB     | 32        | 128          | 3           | 32        | 0.9 | 0.5 | 1.0    |
| POLBLOGS  | 64        | 128          | 1           | 64        | 0.1 | 0.9 | 0.001  |

Table 4: Parameter Setting for Matrix Factorization

| DATASET | DIMENSION | $\ell_2$ NORM WEIGHT | LEARNING RATE | WEIGHT |
|---------|-----------|----------------------|---------------|--------|
| CORA | 32 | 1E-5 | 0.0025 | 1E-5 |
| CITESEER | 64 | 1.0 | 0.01 | 10.0 |
| PUBMED | 128 | 0.001 | 0.0075 | 0.001 |
| WEBKB | 32 | 1.0 | 0.01 | 1.0 |
| POLBLOGS | 128 | 0.001 | 0.005 | 10 |

Table 5: Parameter Setting for Laplacian Eigenmaps

| DATASET | DIMENSION | LEARNING RATE | WEIGHT |
|---------|-----------|---------------|--------|
| CORA | 128 | 0.5 | 10.0 |
| CITESEER | 64 | 0.5 | 10.0 |
| PUBMED | 128 | 0.5 | 10.0 |
| WEBKB | 64 | 0.5 | 10.0 |
| POLBLOGS | 128 | 0.0075 | 1E-6 |

# 5 Analysis of convergence and distortion reduction

We conduct comparison experiments to analyze the convergence of curvature loss and the deduction of distortion (Eq. 1 in the main text of this paper) during optimizing. We run the comparisons ten times, and Figure 2-4 reported the trends (both mean and variance) of curvature loss and the distortion by using LE, MF, and SDNE on Cora dataset, respectively. Figure 2 shows that LE with curvature regularization (blue) get lower curvature loss (solid line) and distortion (dashed line) than the original LE (orange) after iterations. Similar patterns are held in Figure 3 (MF) and Figure 4 (SDNE).

Figure 2: The trends of curvature loss and the distortion for Laplacian Eigenmaps (LE) on the Cora dataset. The $x$ axis denotes the iteration number, the left $y$ axis and right $y$ axis denote the distortion (Eq. 1 in the main text of this paper) and the curvature loss, respectively. LE with curvature regularization (blue) get lower curvature loss (solid line) and distortion (dashed line) than the original LE (orange) after iterations.

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

Table 6: Parameter Setting for Structural Deep Network Embedding

| DATASET | DIMENSION | HIDDEN UNITS (TWO LAYER) | $\alpha$ | $\beta$ | LEARNING RATE | WEIGHT |
|---------|-----------|--------------------------|----------|---------|---------------|--------|
| CORA | 32 | (500,100) | 1 | 10 | 0.0025 | 1 |
| CITESEER | 32 | (100,100) | 1 | 1E-5 | 0.01 | 1 |
| PUBMED | 128 | (200,100) | 1 | 1E-6 | 0.01 | 0.001 |
| WEBKB | 32 | (100,100) | 1 | 0.1 | 0.01 | 0.1 |
| POLBLOGS | 32 | (100,100) | 1 | 1E-5 | 0.01 | 1 |

Figure 3: The trends of curvature loss and the distortion for Matrix Factorization (MF) on the Cora dataset. The $x$ axis denotes the iteration number, the left $y$ axis and right $y$ axis denote the distortion (Eq. 1 in the main text of this paper) and the curvature loss, respectively. MF with curvature regularization (blue) get lower curvature loss (solid line) and distortion (dashed line) than the original MF (orange) after iterations.

Figure 4: The trends of curvature loss and the distortion for SDNE on the Cora dataset. The $x$ axis denotes the iteration number, the left $y$ axis and right $y$ axis denote the distortion (Eq. 1 in the main text of this paper) and the curvature loss, respectively. SDNE with curvature regularization (blue) get lower curvature loss (solid line) and distortion (dashed line) than the original SDNE (orange) after iterations.