[Reviews · NeurIPS 2020]

Review 1

Summary and Contributions: UPD I would like to thank the authors for their response. After reading it carefully, together with the rest of the reviews, I am happy to see that most of my concerns were covered. As a result, I am happy to raise my rating given that the authors will update the final version of the paper accordingly. ==================== This work proposes a novel regularizer for graph representation learning, that takes into consideration the geometry of the representation space. The authors observe that the mainstream graph embedding methods learn an embedding space and then use the Euclidean distances in this space to uncover patterns and tackle classification tasks. However, there's no clear constraints about how the geometry of the embedding space within the Euclidean space. Filling this gap, the authors hypothesize that by biasing the embedding algorithms towards flat embedding surfaces, the final embeddings will be more informative when combined with the Euclidean distance measure. The work defines a measure of curvature and provides a framework to modify existing embedding methods and add the curvature as a regularizer. The experiments show that indeed the performance of these curvature-regularized embeddings on node classification and link prediction tasks is superior compared to before.

Strengths: * With this work, the authors revisit a common assumption that "similar nodes should be closed to each other in the embedding space" and examine what "close" means. Connecting the works on representation learning with literature from differential geometry sounds very promising, especially when the reason that these representation are so powerful is the geometry of the learnt space. * The main contribution of the work is methodological; the authors describe how one can change potentially any algorithm for graph representation learning to consider and regularize for the curvature of the end result. This has quite broad impact.

Weaknesses: * Although the authors try to position the paper in a way that it builds on intuitive observations, the presentation could still be improved. * The novelty of the work is relatively limited, especially for a venue like NeurIPS. * The experimental sections feels quite dry; it just serves as a poster where the authors showcase numbers. There are no real learning for the reader apart from pointing out that the method works. I would have expected to learn something more from that section.

Correctness: Overall, both the theoretical claims and the empirical methodology appears to be correct.

Clarity: The authors' approach in presenting the paper is aiming at building a strong intuition on why the observations and the claims that the works is based on make sense. This is very commendable, especially since the work is heavily based on geometry. However, there is still room for improvement and there are missed opportunities, e.g., what I mentioned in the weaknesses regarding the experimental section. I am including some suggestions below.

Relation to Prior Work: Overall, the authors do a reasonable presentation of the prior work. There are connection to the graph mining literature that are overseen (I provide more context below).

Reproducibility: Yes

Additional Feedback: * One philosophical question that comes to mind when reading the three observations in the Introduction and while going over the example in Figure 1 is the following: Could it be that the exact reason that the representation methods learn interesting patterns is the fact that they are allowed to twist and curve the space as required, in order to bring nodes that are far apart in terms of graph distance close together in terms of Euclidean distance? It is not obvious to me that constraining the optimization algorithm of this ability can only have positive outcome. There's a parallel to be drawn here with the kernel trick in Support Vector Machines, where we are allowed to embed the data in a higher dimension, where the classes become linearly separable. * In Definition 2, the notation could be simplified by denoting \Gamma_{i, j} to be the set of edges {(u, v)} along the shortest path from i to j. This way, the sum could be defined over (q', q'') \in \Gamma_{i, j} * The sectional curvature could have been more thoroughly introduced. Unfortunately, for a paper that is heavily based on geometric notions, there's a clear shortage of pictures. What better way to build intuition for geometric properties than a plot or two? * The way that ABS is defined, it's clear that the participation of each node q to the final cost depends on how many shortest paths the node participates in. I.e., K_q is more "important" for a node that participates in bridge-like edges. This notion has been explored deeply in the node centrality literature, through the notion of betweenness centrality. It would be interesting to see a direct connection between this and the ideas presented in the paper here. * As a direct corollary of the above, in the Section 4.2 where the authors explore sampling methods, it would be interesting to look at related works that attempt to approximate the betweenness centrality computation through sampling. Similarly, the authors could look into using the PageRank of nodes as a way to inform sampling for the random-walk based curvature regularization. * One question that came to mind when reading through the experimental Section was how bad the curvature was in the tested methods. The authors only provide performance results. However, it would be interesting to see how much worse was the curvature loss in the different setups. * One claim that authors make early on is that this work focuses on graphs with no features on the nodes/edges. Given that graphs with features (or even heterogeneous graphs) are becoming more and more prevalent, and there are more and more fine-tuned methods that take advantage of this information, it would be interesting to see if and how they think that their contribution can be applied to these cases. Especially since, in the cases where there's more than the structural information that the embedding tries to capture, allowing for curvature feels like a way to encode these non-structural properties. * Minor typos sprinkled throughout the text. E.g.: line 117: consider line 130: alone line 143 : it measure line 150: intersection of each the plane line 169: by an signed Table 2: the (LE, Polb) number is bold by mistake


Review 2

Summary and Contributions: [POST REBUTTAL] I thank the authors for the detailed response, which address all of my concerns. I carefully check the proofs in the supplementary materials, and am convinced. Hence I raised my score. ---------------------------------------------------------------------------------------------------------- In this work, a novel regularization method is proposed to reduce the pattern distortion of the embedding manifold in the context of graph embedding. In particular, the authors formulate the pattern distortion concept and proposed the "angle-based sectional curvature (ABS curvature)", and proved that by decreasing the curvature, the embedding manifold would have less distortion. These formulations are novel. Moreover, the authors developed proxy regularizers induced from the ABS curvature and proposed a two-phase optimization method utilizing the proposed regularizer.

Strengths: Theoretically, the authors observed that the distortion pattern of the embedding manifold is a commonly encountered problem and causes underperformance of downstream machine learning tasks. Moreover, novel concepts including the ABC curvature. Experimentally, the authors performed extensive experiments evaluating the proposed method on various datasets and showed promising results.

Weaknesses: The major concern is that the statement of the theorem could be more clarified. In particular, the conditions of Theorem 1 in the supplementary materials, where the proof is included, is different from that in the main paper. Moreover, in the statement of Theorem 1, the meaning of "the 2-dimensional subspace of embedding manifold M" is not clear, and not explained in the context. As I take it, the condition imposed on the 2-dimensional subspace should be true for ANY 2-dimensional subspace of the embedding space. If this is the case, this introduces the computational burden for large-scale problems, which limits the efficacy of the proposed method. In the optimization algorithm (algorithm 1), the authors make no mention of doing the "2-dimensional subspace" check. A minor concern is a lack of theoretical backgrounds for the variants of \Omega_c, i.e. \Omega_s and \Omega_a. More discussion is needed on how well these regularizers approximate the original one. Another minor concern relates to the definition of distortion (Eq.1), which evaluates the difference between the geodesic distance and Euclidean distance of two nodes. The proposed method aims to decrease this distortion. However, how this property yields a good manifold representation of the graph is not studied, given the chance that the graph has complicated topology or has a hyperbolic underlying structure.

Correctness: The claims and method are correct. The empirical methodology involving the original ABS curvature, need to be further improved. The concept of discrete sectional curvature need to be improved.

Clarity: The claims and methods are correct. The empirical methodology involving the original ABS curvature is well theoretically grounded but might not scale to large-scale problems. The variant methods, which are based on the variants of the original regularizer, lack solid theoretic grounds or ablation studies empirically.

Relation to Prior Work: The authors clearly state their contribution in the paper.

Reproducibility: Yes

Additional Feedback: This paper could be improved by including more discussion on the justification of distortion and variants of \Omega_c, either theoretically or empirically or both. Moreover, the assumptions in Theorem 1 should be further clarified.


Review 3

Summary and Contributions: This paper raises and formulates a new problem in proximity-preserving graph embedding, graph topology patterns that are preserved well into an embedding manifold, may distort in the ambient embedding Euclidean space, and hence to detect them becomes difficult for machine learning models operating in Euclidean space. As the distortion is caused by curved embedding manifolds, this paper proposes to prevent the distortion by flattening the embedding manifolds. The authors present a novel angle-based sectional curvature, termed ABS curvature, as well as three kinds of curvature regularization to induce flat embedding manifold in proximity-preserving graph embedding. The curvature regularizations are integrated into some embedding methods and evaluated on real graph datasets. The experimental results show the proposed curvature regularizations improve the performance of embedding methods on node classification and link prediction tasks.

Strengths: S1: The raised problem in this paper is important but neglected in proximity-preserving graph embedding, which may attract attention from graph embedding community. S2: This paper proposes a new curvature, ABS-curvature, to measure how far an embedding manifold is from being flat. S3: Based on ABS-curvature, this paper designs three kinds of curvature regularization which can be integrated into existing embedding methods. S4: This paper provides a new perspective to view the negative sampling strategy in graph embedding. S5: Experimental results show the proposed curvature regularization benefits existing graph embedding methods in some real graph datasets on node classification and link prediction tasks.

Weaknesses: W1: It seems that the proposed curvature regularizations are difficult to optimize. How is the convergence of Algorithm 1? The author should provide theoretical or empirical results to demonstrate its convergence. W2: The criterion to evaluate the model performance in the experiment is not clear. What is mean prediction precision in the link prediction task? W3: The time complexity of the original curvature regularization is too high to apply to real large graphs.

Correctness: The method and empirical methodology are correct for me. However, there is a slight difference between the theoretical results (Lemma 1 and Theorem) in the main text and the appendix. The theoretical results in the appendix seem correct.

Clarity: The paper is well-written and easy to follow.

Relation to Prior Work: This paper elaborates on the main difference from previous works.

Reproducibility: Yes

Additional Feedback: The paper raises an interesting problem in graph embedding, and proposed novel curvature regularizations to address the problem. The proposed regularizations are validated in 8 real graph datasets. I have some additional concerns and suggestions about this paper. 1. Do the proposed curvature regularizations prevent the distortion defined in Eq. 1 in experiments? The authors should validate it by calculating the distortion $\rho$ for the embeddings in experiments. 2. The convergence of Algorithm 1 is not clear. It would be nice if the authors can present convergence curve in experiments on real graph datasets. 3. In literature, there are many manifold learning (non-linear dimension reduction) technologies. Can we prevent the distortion of embedding manifold via applying those technologies to the curved graph embedding? What's the difference between this strategy and the curvature regularization? 4. The authors are suggested to share the source code to the public. i have read the rebuttal, thanks


Review 4

Summary and Contributions: This paper draws attention to the distortion problem in proximity-preserving graph embedding caused by curved embedding manifolds, which is crucial but neglected. Based on several observations of causes of the distortion and the impact of the distortion, this paper proposes a novel angle-based sectional curvature to measure the curvature in the embedding manifold and several curvature regularizations to alleviate the distortions and enforce the embedding manifolds to be flat in Euclidean space. Popular node embedding methods equipped with the proposed angle-based sectional curvature regularization achieves superior improvements on several datasets. Experimental results indicate the importance of the distortion in graph embedding and demonstrate the impacts of the proposed curvature regularization method.

Strengths: 1.This paper provides solid theoretical analysis of the distortion and curvature in proximity-preserving graph embedding. 2.This paper proposes a novel measure ABS curvature and a curvature regularization to cope with the distortion to induce a flat embedding manifold. 3.The proposed regularization term performs well and significantly improve several graph embedding methods (Matrix Factorization, DeepWalk and etc.) on various datasets. The general regularization term can be applied to more methods to induce embedding manifolds with low distortion. 4.Theoretical analysis and solutions are formulated well, which is clear to understand .

Weaknesses: This paper is well-written, methods are straightforward, and the experiments are demonstrative but the length of the paper exceeds 8.

Correctness: Claims and methods in this paper are correct. This paper points out the distortion of embedding manifolds in proximity-preserving graph embedding. To make the embedding manifolds flat, it provides curvature regularizations to restrict the curvature when embedding.

Clarity: Yes, the paper is well written and I can clearly understand the proposed method.

Relation to Prior Work: Yes, this paper clearly discussed the difference. This paper discovers the curvature and distortion problem in graph embedding and presents a novel method to solve it.

Reproducibility: No

Additional Feedback: I've read all reviews and the response and it covers most concerns. The idea of this paper is novel and effective. Some mistakes and unclear statements should be revised. I keep my rating.

[Author Response · NeurIPS 2020]

We thank all the reviewers for your thoughtful feedback! We will incorporate our responses into the paper.

*–To Reviewer #1 & #3–* **Q1** & **Q7:** *To curve the embedding space is required for obtaining good graph embeddings, like*
*kernel tricks in SVM. The proposed regularizations to reduce curvature may adversely affect on the graph embedding.*
**Response:** We partially agree with your opinion. Curving an embedding space is

indeed required, where a useful curvature is of benefit to preserving graph topology.
For instance, the green dashed circles in Fig. 1 (from Fig. 1 in paper) indicate
useful curvatures that form good distributions, where the connected nodes are close
and disconnected nodes are far apart. Hyperbolic space is also an example, whose
useful negative curvature is suitable for embedding tree-structured graphs.

Figure 1: Useful and useless curvatures.

However, useless or harmful curvatures are also inevitable in proximity-
preserving embedding because the existing strategies have no restriction on the
"non-local" curves of an embedding space, such as the global "swirling" curve
in Fig. 1B1. Such harmful curvatures bring undesired distortions because unsupervised graph embedding aims to
faithfully preserve graph topology patterns into an embedding space. In other words, any changes/distortions in patterns
is not desired, which is fundamentally different from kernel tricks in SVM, which optimize distribution pattern (by
maximizing separation) *controlled by* data labels in a supervised manner.
We stress that the proposed method differentiates useful from harmful curvatures because the former will contribute
to the reconstruction term in the objective function while the latter will not and thus be eliminated. In Fig. 1B2, to
illustrate a desired "oracle", the global swirling curvature is reduced while the green-circle curvature still exists.

**Q3:** *A more thorough introduction to sectional curvature, especially using figures.*
**Response:** Thanks for the suggestion. As sectional curvature is defined in the space of more

than 3 dimensions, it's hard to visualize in a figure. To provide similar intuition, in a new figure
(Fig. 2), we will instead illustrate normal curvature, which can be considered as a degenerate
case of sectional curvature in a 3-dimension space. We will add a more thorough introduction
of sectional curvature in our revised version.

**Q4:** *Connection between the curvature regularization and betweenness centrality.*
**Response:** Your finding is absolutely right. The proposed regularization constrains more

Figure 2: Normal curvature.

on nodes with high betweenness than others– which we think makes sense to prevent distortion: A node with high
betweenness in a graph is corresponding to a "bottleneck" region in an embedding manifold, i.e., a small region where
lots of geodesic curves pass through. Taking Fig. 1B2 as an example again, the nodes in the green dashed circle
have high betweenness. If we curve or fold the embedding space around these nodes, the Euclidean distances of most
node pairs will be changed, which will bring large distortions in terms of Eq. 1 (the divergence between the geodesic
and Euclidean metric) in paper. To contrast, if we curve the embedding space around the pink dashed circle (low
betweenness), it brings limited distortion. Thus, indeed, high-betweenness nodes should be constrained more.

**Q5:** *Betweenness and PageRank can be used to inform sampling for the two efficient variants of curvature regularization.*
**Response:** Thanks for your constructive suggestion. Indeed, our sampled regularization variant $\Omega_s$ is equivalent to the
betweenness-informed strategy you suggested. It samples shortest paths between nodes, by which nodes with high
betweenness naturally have a high probability to be sampled. We believe a PageRank-based method is also meaningful,
which we will discuss and experiment in our final version.

**Q6:** *To see how much worse was the curvature loss in the different setups.*
**Response:** Great suggestion. We now conducted a comparison experiment to illustrate

the curvature loss by using Laplacian Eigenmaps (LE) on the Cora dataset. We ran the
comparison 10 times, and Fig. 3 reported the mean and variance. One can see that LE with
curvature regularization got lower curvature loss (solid line) and distortion (dashed line)
than the original LE after iterations. We will add more extensive/systematic comparisons
to our final version.

*–To Reviewer #2 & #3–* **Major concern 1:** *Different statements of Theorem 1.*

Figure 3: Curvature loss and distortion.

**Response:** Sorry for the mistake. We gave the wrong conditions in the main paper and
corrected it in the supplementary material. We will correct the conditions of Theorem 1 in our revised version.
**Major concern 2:** *Does the condition in Theorem 1 apply to any 2-dimensional subspace of the embedding space? If*
*so, it may introduce a large computational burden during the optimization, which does not mention in the Algorithm 1.*
**Response:** While the condition in Theorem 1 applies to any 2-dimensional subspace, we stress that it is only for theo-
retical proof and does *not* introduce computational overhead. The 2-dimensional subspaces are only used theoretically
in Theorem 1 to prove the proportional relation. In contrast, Algorithm 1 optimizes directly node embeddings in the
overall high-dimensional space, rather than every 2-dimensional subspace.

*–To Reviewer #4–* Formatting issue: We appreciate your recognition of our work. We will correct our paper to meet the
format requirement of NeurIPS.

[Meta-Review · NeurIPS 2020]

The paper was reviewed by 4 expert reviewers who really appreciated the paper and, even more importantly, the rebuttal, which they felt answered their concerns. The paper is therefore accepted; the authors should consult the reviews for suggestions for the camera ready submission.